# Regulation of Genes Related to Cognition after tDCS in an Intermittent Hypoxic Brain Injury Rat Model

**DOI:** 10.3390/genes13101824

**Published:** 2022-10-09

**Authors:** Jin-Won Lee, Won-Hyeong Jeong, Eun-Jong Kim, Insung Choi, Min-Keun Song

**Affiliations:** 1Department of Physical & Rehabilitation Medicine, Chonnam National University Hospital, 42 Jebong-ro, Dong-gu, Gwangju 61479, Korea; 2Department of Physical & Rehabilitation Medicine, Chonnam National University Medical School, 160, Baekseo-Ro, Dong-Gu, Gwangju 61469, Korea

**Keywords:** hypoxic brain injury, transcranial direct current stimulation, gene regulation

## Abstract

**Background**: Hypoxic brain injury is a condition caused by restricted oxygen supply to the brain. Several studies have reported cognitive decline, particularly in spatial memory, after exposure to intermittent hypoxia (IH). However, the effect and mechanism of action of IH exposure on cognition have not been evaluated by analyzing gene expression after transcranial direct current stimulation (tDCS). Hence, the purpose of this study was to investigate the effects of tDCS on gene regulation and cognition in a rat model of IH-induced brain injury. **Methods**: Twenty-four 10-week-old male Sprague–Dawley rats were divided into two groups: IH exposed rats with no stimulation and IH-exposed rats that received tDCS. All rats were exposed to a hypoxic chamber containing 10% oxygen for twelve hours a day for five days. The stimulation group received tDCS at an intensity of 200 µA over the frontal bregma areas for 30 min each day for a week. As a behavior test, the escape latency on the Morris water maze (MWM) test was measured to assess spatial memory before and after stimulation. After seven days of stimulation, gene microarray analysis was conducted with a KEGG mapper tool. **Results**: Although there were no significant differences between the groups before and after stimulation, there was a significant effect of time and a significant time × group interaction on escape latency. In the microarray analysis, significant fold changes in 12 genes related to neurogenesis were found in the stimulation group after tDCS (*p* < 0.05, fold change > 2 times, the average of the normalized read count (RC) > 6 times). The highly upregulated genes in the stimulation group after tDCS were SOS, Raf, PI3K, Rac1, IRAK, and Bax. The highly downregulated genes in the stimulation group after tDCS were CHK, Crk, Rap1, p38, Ras, and NF-*k*B. **Conclusion**: In this study, we confirmed that SOS, Raf, PI3K, Rac1, IRAK, and Bax were upregulated and that CHK, Crk, Rap1, p38, Ras, and NF-*k*B were downregulated in a rat model of IH-induced brain injury after application of tDCS.

## 1. Introduction

Approximately 30–50% of children with hypoxic brain injury are known to have developmental delays accompanied by neurologic symptoms [1], and patients who experience severe bleeding during cardiac or aortic surgery also experience hypoxic–ischemic brain damage and neurological decline even after recovery [2].

Exposure to intermittent hypoxia (IH) exposure in obstructive sleep apnea syndrome may cause cognitive decline due to apoptosis of neurons in the cortex and hippocampus [3,4,5]. Several studies have reported cognitive decline, particularly in spatial memory, after IH exposure [6,7,8,9,10].

Transcranial direct current stimulation (tDCS) may facilitate cortical neuroplasticity [11]. One of the mechanisms of the effect of tDCS is to modulate cortical excitability by reducing GABA and glutamatergic neuronal activity [12]. However, the effect of tDCS on cognition is debated. Several studies have shown that tDCS is effective in improving various cognitive functions in aged people [13,14]. However, there are a few reports describing the neutral or negative effects of tDCS on cognition [15,16]. Some clinical studies have reported that tDCS is ineffective at improving memory and executive function [17,18].

The effect and mechanism of action of tDCS on cognition have not been evaluated by analyzing gene expression after tDCS in IH-induced brain injury. The purpose of this study was to investigate the effect of tDCS on gene regulation and cognition in a rat model of IH-induced brain injury.

## 2. Materials and Methods

### 2.1. Experimental Subjects

The experimental subjects were 10-week-old male Sprague-Dawley rats (Samtako Co., Osan, Korea) weighing 300 ± 50 g. All subjects were housed under regular circumstances in the University Animal Care Laboratory. The study protocol was approved by the Institutional Animal Ethics of University Animal Care and Committee (CNUH IACUC-18018), and all experimental procedures followed the guidelines of the IACUC.

A total of 24 rats were randomly separated into two groups: a control group (n = 12) and a stimulation group (n = 12). After being exposed to 12 h/day in a hypoxic chamber with a 10% oxygen concentration for five days, the control rats were exposed at normal oxygen concentrations to compare their spontaneous recovery and application of tDCS.

### 2.2. Methods

The study was sequentially performed as scheduled (Figure 1). Twenty-four 10-week-old rats were subjected to acclimatization for three days. Next, pretraining with the Morris water maze (MWM) was performed for three consecutive days. Hypoxic brain injury was induced over five days. tDCS stimulation was conducted the day after the end of hypoxic brain injury induction. tDCS was applied for seven days. All the rats were sacrificed the day after the end of the experiment, and hippocampal tissues were extracted for RNA sequencing.

#### 2.2.1. Hypoxic Brain Injury Rat Model

The rat model of hypoxic brain injury was induced with a hypoxic chamber (Figure 2). Animals rested for 12 h/day (n = 24) in one identical commercially designed chamber (30 × 320 × 320 inches) for five days under conditions with 10% oxygen concentration. Deviations from the desired oxygen concentration were corrected by the addition of N_2_ through solenoid valves. The humidity was measured and maintained at 40–50% by circulating gas through the freezer and using silica gel. The ambient temperature was kept at 22–24 °C [19].

#### 2.2.2. Transcranial Direct Current Stimulation

For tDCS, we used a battery-driven, constant-current stimulator (HDC manufactured by Newronika s.r.l., Italy, and distributed by Magstim Co. Ltd., Whitland, Wales, UK). For the two-channel anodal method, cup-shaped active electrodes (1 cm × 1 cm) were placed on the frontal bregma area; in contrast, for the cathodal method, a 0.5-cm sponge pad was applied to the neck (Figure 2). Electrical stimulation was applied at an intensity of 200 μA for 30 min over a period of seven consecutive days.

#### 2.2.3. Neurocognitive Behavioral Test

Evaluation of spatial learning and memory was assessed through the MWM test developed by Morris et al. [20]. All the methods for the MWM test were performed according to the methods of a previous study [21]. The tests were conducted in a circular pool with a diameter of 184 cm and a height of 60 cm. The pool was filled with water and maintained at 22 ± 2 °C. The pool was virtually divided into four quadrants, and one quadrant was set as the target. Visual symbols were assigned to the perimeter of each quadrant. A circular escape platform (diameter, 10 cm; height, 38 cm) was positioned in the center of the target quadrant. A platform was submerged one centimeter below the water level.

All groups underwent pretraining for three consecutive days before inducing hypoxic brain injury. The animals were randomly placed in the water maze facing the maze wall entry points and distributed evenly around the perimeter of the maze. After finding the platform, the rats stayed there for 10 s until the next experiment. If the rat could not find the hidden platform within 120 s, the rat was placed on the platform for 15 s so that it could recognize the location of the platform. The rat was displaced from the pool and placed back in its cage for five minutes. Then, the second trial was performed.

The MWM test was performed to evaluate spatial memory on the day after stimulation. The rats tried to find the platform below the surface of the water within 300 s. The escape latency (time taken to reach the platform) was automatically calculated by an Ethovision Color-Pro^®^ video tracking system (Nodulus, Wageningen, The Netherlands).

#### 2.2.4. RNA Sequencing Analysis

All the methods of RNA sequencing were performed according to the methods of a previous study [22]. After sacrifice, hippocampal tissues from all rats were extracted for RNA sequencing.

##### RNA Isolation

Total RNA was isolated using TRIzol reagent (Invitrogen, Waltham, MA, USA). Assessment of RNA quality was performed with an Agilent 2100 bioanalyzer using the RNA 6000 Nano Chip (Agilent Technologies, Amstelveen, The Netherlands), and RNA quantification was performed using an ND-2000 Spectrophotometer (Thermo Inc., Waltham, MA, USA).

##### Library Preparation and Sequencing

For control and test RNAs, library construction was performed according to the manufacturer’s instructions using QuantSeq 3′ mRNA-Seq Library Prep Kit (Lexogen, Inc., Wien, Austria). In summary, 500 ng of total RNA was prepared, oligo-dT primers containing an Illumina-compatible sequence at the 5′ end were hybridized to the RNA, and reverse transcription was performed. The second-strand synthesis was started using random primers with an Illumina-compatible linker sequence at the 5′ end after the degradation of the RNA template. The double-stranded libraries were purified using magnetic beads to remove all reaction components. The library was amplified to add the full adapter sequences required for cluster generation. The finished library was purified from PCR components. High-throughput sequencing was performed as single-ended 75 sequencing using NextSeq 500 (Illumina, Inc., San Diego, CA, USA).

##### Data Analysis

QuantSeq 3′ mRNA-Seq reads were aligned using Bowtie2 [23]. Bowtie2 indices were generated from genome assembly sequences or representative transcript sequences for alignment to the genome and transcriptome. The alignment files were used to assemble transcripts, estimate their amounts, and detect differential expression of genes. Differentially expressed genes were determined based on the counts of unique and multiple alignments using Bedtools’ coverage [24]. Read count (RC) data were processed according to the Quantile normalization method using EdgeR within R using Bioconductor [25]. Gene classification was set using the DAVID (the Database for Annotation, Visualization, and Integrated Discovery, https://david.ncifcrf.gov/, accessed at 26 October 2021) and Medline databases. Then, the neurotrophin signaling pathway was elicited with the KEGG mapper–Search & Color Pathway (http://www.genome.jp/kegg/tool/map_pathway2.html, accessed at 26 October 2021) to put the Entrez ID and input data.

## 3. Statistical Analysis

The sample size of this study was calculated according to Cohen’s formula [26], and the effect size was set to 0.95 based on the large-sized F-value. The number of groups was set to 2, the number of measures was set to 2, the effect size was set to 0.95, the significance level was set to 0.05, and the statistical power was set to 0.70. The total sample size required for repeated-measures ANOVA (interaction of time with the group), calculated using G * power [27], was 24.

All statistical analyses were performed using SPSS for Windows (version 26.0; Chicago, IL, USA), and all data are presented as the mean ± standard deviation (SD). Escape latency and velocity were analyzed by one-way ANOVA, repeated-measures ANOVA, and subsequent post hoc tests.

## 4. Results

### 4.1. Neurocognitive Behavioral Test

The escape latency on the MWM test before tDCS stimulation was 93.33 ± 38.92 s and 104.26 ± 21.53 s for the control and stimulation groups. There was no significant difference between the two groups before tDCS stimulation (*p* = 0.404). The escape latency on the MWM test after tDCS stimulation was 79.92 ± 36.61 s and 50.94 ± 37.07 s for the control and stimulation groups, respectively. There was no significant difference between the two groups after tDCS stimulation (*p* = 0.067). The escape latency on the MWM test was also analyzed by ANOVA with test time as a repeated measure. The results showed a significant change over time (F = 12.451, *p* = 0.002) and a significant time × group interaction (F = 4.452, *p* = 0.046) (Figure 3). tDCS has an effect on cognitive functioning, as demonstrated by statistically significant changes in escape latency and the time differences between the stimulation and control groups.

### 4.2. RNA Sequencing Data

QuantSeq RNA analysis revealed 17,312 gene symbols. Then, symbols with significant fold changes in the stimulation group compared with the control group were identified. Significant fold changes of 180 genes were shown in the tDCS stimulation group (*p* < 0.05, fold change > 2 times, average of normalized read counts (RCs) > 6 times) (Table 1). Whole sequencing data was included in Appendix A.

### 4.3. KEGG Mapper Tool Analysis

After tDCS, neurotrophin signaling pathways were analyzed with a KEGG mapper tool. Compared with the control group, the upregulated genes in the stimulation group after tDCS were SOS, Raf, PI3K, Rac1, IRAK, and Bax (*p* < 0.05). Compared with the control group, the downregulated genes in the stimulation group after tDCS were CHK, Crk, Rap1, p38, Ras, and NF-kB (*p* < 0.05) (Figure 4).

## 5. Discussion

In this study, RNA sequence analysis was performed after tDCS stimulation in a rat model of IH-induced injury. In a previous study, a neurogenesis induction effect of tDCS after experimental stroke was observed at the cellular level [28]. To examine the expression changes in the transcriptome related to this effect, the neurotrophin signaling pathway, which is related to neurogenesis, was selected from among the enriched pathways in the KEGG analysis. Among the genes corresponding to this pathway, six genes (SOS, Raf, PI3K, Rac1, IRAK, and Bax) were upregulated, and six genes (CHK, Crk, Rap1, p38, Ras, and NF-kB) were downregulated.

The neurotrophin signaling pathway is a pathway activated by neurotrophin, a protein that controls the function of neurons in many ways. Four types of neurotrophin are known, which are nerve growth factor (NGF), brain-derived neurotrophic factor (BDNF), neurotrophin-3 (NT-3), and neurotrophin-4 (NT-4). These neurotrophins are essential for maintaining the survival, morphology, and differentiation of normal neurons, and have various roles, such as synaptic function control and plasticity control. This pathway consists of two types of receptors: tropomyosin-related kinase (Trk) receptor and p75 neurotrophin receptor (p75NTR). When neurotrophin binds to these receptors, each downstream pathway is activated [29].

The Trk receptor-mediated pathway almost always promotes neuronal survival and differentiation, and there are three major intracellular signaling pathways. The first is the mitogen-activated protein (MAP) kinase cascade by Ras activation, which is a pathway that promotes neuronal differentiation. In this study, Son of Sevenless (SOS) and Raf were found to be upregulated among the genes involved in this pathway, but Ras and p38 were downregulated. SOS acts as a Ras exchange factor, and Raf phosphorylates Mek1 and Mek2 to phosphorylate and activate Erk1 and Erk2. Furthermore, Ras plays an important role in promoting neuronal differentiation by stimulating signaling of the c-Raf-Erk, p38MAP kinase, and class I phosphatidyl inositol-3 (PI3) kinase pathways. P38MAP kinase is responsible for the phosphorylation of cyclic AMP response element binding protein (CREB) by activating MAP kinase-activated protein kinase-2 (MK-2).

Interestingly, among the intermediate genes of this MAPK pathway, SOS and Raf were upregulated, but Ras and p38 were downregulated. This finding suggests that tDCS affects neuronal differentiation in IH-induced injury, but to confirm this effect, further studies including immunohistochemical analyses should be performed by selecting candidate genes coding for the neuronal differentiation process. In a previous study, cathodal tDCS was found to induce upregulation of osteopontin (OPN) [10], which is known to increase neuronal differentiation of neural stem cells after cerebral ischemia [30]. In this study, by applying anodal tDCS after exposure to IH, the MAPK pathway was found to play an important role as another tDCS-regulated pathway that affects neuronal differentiation.

On the other hand, among the genes in the Crk-C3G-Rap1 signaling pathway, which is a minor pathway that sustains the activation of MAPK, Crk, and Rap1 were downregulated. The result that both genes were downregulated is not in the same direction as our previous hypothesis, but tDCS may act to inhibit the MAPK cascade in IH conditions. 

The second Trk receptor-mediated pathway is the PI3 kinase pathway, which promotes neuronal survival. Among the genes in this pathway, PI3K was upregulated, but CHK and NF-kB were downregulated. When PI3K is activated, Akt is activated along the lower signaling pathway, and eventually, IkB is degraded to release NF-kB, which promotes neuronal survival. CHK acts as a signaling molecule that is recruited to the Trk receptor. PI3K, CHK, and NF-kB showed regulation in the opposite direction; furthermore, in the abovementioned upregulation of OPN by cathodal tDCS [10], OPN also enhanced the survival of neural stem cells [30]. Therefore, the effect of anodal tDCS on the PI3K pathway and neuronal survival also needs to be further studied.

The third Trk receptor-mediated pathway is the phospholipase C-r1 (PLC-r1) pathway that promotes synaptic plasticity. In the results of this study, there were no significantly regulated genes belonging to this pathway. This may be related to the fact that, unlike other pathways mediated by the Trk receptor, the PLC-r1 pathway is considered to have undergone adaptation to be integrated into the Trk receptor during the evolution process [31].

The p75NTR-mediated pathway promotes neuronal apoptosis, and there are several major intracellular signaling pathways. One of them is the Jun kinase pathway, and signaling of this pathway leads to p53 activation and apoptosis. In the results of this study, Rac1 and Bax were upregulated among the genes involved in this pathway. Rac1 plays an essential role in p75NTR-mediated apoptosis, particularly in oligodendrocytes [32]. Bax is a pro-apoptotic gene activated by p53. From the results of the upregulation of these two genes, it can be expected that tDCS in IH conditions would have the effect of promoting apoptosis through JNK signaling. Previous studies have reported that tDCS influences the apoptotic process. In ischemic mice, cathodal tDCS reduced the number of caspase-3-positive cells, which represent apoptotic cells, but anodal stimulation increased the same [33]. Anti-apoptotic proteins have been reported to be upregulated in fibroblasts exposed to electrical fields in vitro [34].

Another pathway is the NF-kB pathway, which induces neuronal survival. Among the genes of this pathway, interleukin-1 receptor-associated kinase (IRAK) was upregulated, but NF-kB was downregulated. IRAK is recruited to the complex formed by Traf6 and p75NTR to phosphorylate IkB and release NF-kB. IRAK and NF-kB also showed conflicting regulation; thus, further research is needed.

According to a previous study, the gene expression pattern and magnitude of the response depend on the tDCS current intensity [35]. Therefore, further studies, including tDCS stimulation with various current intensities, are needed for genes belonging to the neurotrophin signaling pathway, including the MAPK, PI3K, and NF-kB pathways, which showed contradictory changes in gene regulation in this study.

However, this study had a small effect size, and the findings do not apply to humans as it was an animal study. Other limitations were the lack of a sham stimulation group for applying tDCS and the lack of quantitative data for selected genes, such as real-time PCR or western blot analysis, which was not supported. Further studies are needed to identify the effective therapeutic intensity of tDCS that may enhance neuroplasticity in irreversible hypoxic brain injury.

## 6. Conclusions

After the tDCS experiment, significant fold changes in 12 genes related to neurogenesis in rats with IH-induced brain injury after tDCS were shown. Therefore, regulated gene biomarkers related to cognition may be helpful in predicting the effect of tDCS in rats with IH-induced brain injury.

## Figures and Tables

**Figure 1 genes-13-01824-f001:**
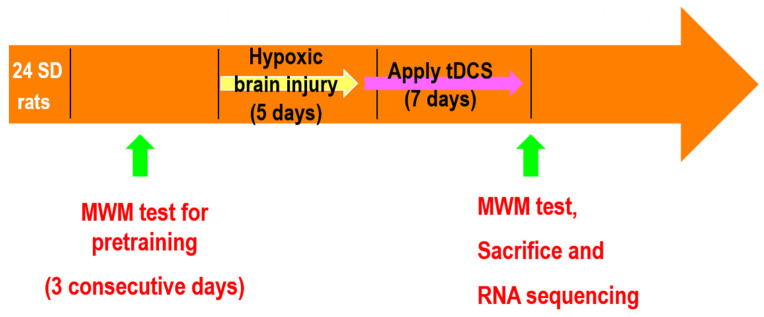
Schematic representation of the experiment.

**Figure 2 genes-13-01824-f002:**
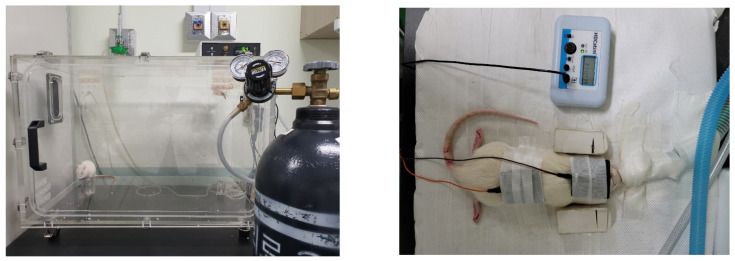
Modeling hypoxic brain injury in a rat with a hypoxic chamber and transcranial direct current stimulation (tDCS). A cup-shaped active electrode (1 cm × 1 cm) was placed on the frontal bregma area. For the cathodal method, a 0.5 cm sponge pad was applied to the neck. Electrical stimulation was applied at an intensity of 200 μA for 30 min over a period of seven consecutive days.

**Figure 3 genes-13-01824-f003:**
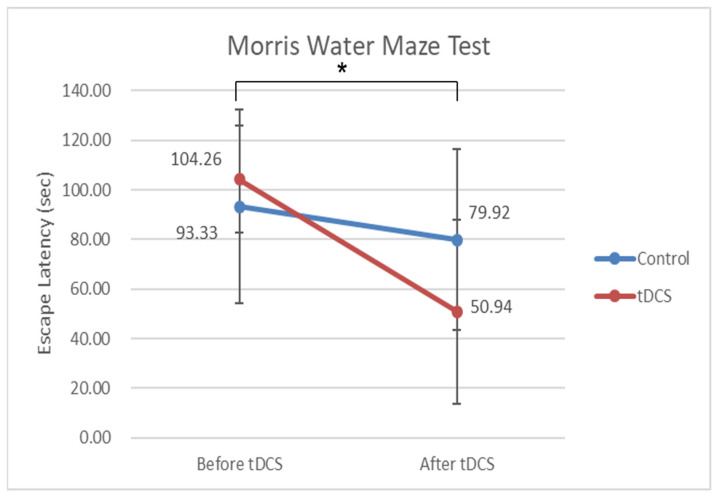
Escape latency on the hidden platform trial on the Morris-water maze (MWM) test. The escape latency on the MWM test before tDCS stimulation was 93.33 ± 38.92 s and 104.26 ± 21.53 s in the control and stimulation groups, respectively. The escape latency on the MWM test after tDCS stimulation was 79.92 ± 36.61 s and 50.94 ± 37.07 s in the control and stimulation groups, respectively. There were no significant differences between the two groups before and after tDCS stimulation (*p* = 0.404, 0.067). However, the results showed a significant change over time (F = 12.451, *p* = 0.002) and a significant time × group interaction (F = 4.452, *p* = 0.046). As evidenced by the statistically significant changes in escape latency and time difference between the two groups, tDCS may affect cognitive function. * *p* < 0.05.

**Figure 4 genes-13-01824-f004:**
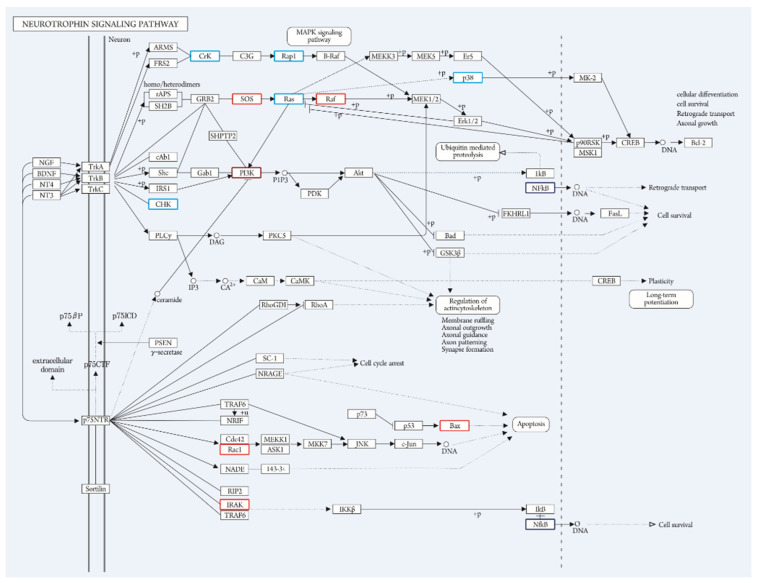
KEGG Mapper Tool Analysis. This figure represents the neurotrophin signaling pathway after tDCS (coral: upregulation, blue: downregulation). Compared with the control group, the upregulated genes in the stimulation group after tDCS were SOS, Raf, PI3K, Rac1, IRAK, and Bax (*p* < 0.05). Compared with the control group, the downregulated genes in the stimulation group after tDCS were CHK, Crk, Rap1, p38, Ras, and NF-kB (*p* < 0.05).

**Table 1 genes-13-01824-t001:** Fold changes in 180 genes with significant fold changes.

ID	Gene Symbol	Fold Change (Stimulation/Control)	Average Normalized Read Counts	
			Control	Stimulation
212	Actr10	2.025	5.673	6.692
216	Actr3	2.259	6.008	7.183
317	Adh5	2.318	4.889	6.102
331	Ado	5.946	3.483	6.055
358	Aes	0.446	8.412	7.247
867	Arl2	0.187	6.446	4.028
879	Arl6ip5	0.416	6.667	5.401
910	Arpp19	4.004	6.677	8.679
1019	Atg3	2.216	4.959	6.106
1038	Atox1	0.090	6.265	2.796
1052	Atp1b2	0.454	6.276	5.138
1094	Atp6v0e2	0.328	7.201	5.595
1127	Aurkaip1	2.759	4.729	6.193
1204	Basp1	0.444	9.225	8.054
1228	Bcas1	0.466	7.632	6.531
1439	Bud31	0.472	6.135	5.053
1453	C1ql3	0.418	6.256	4.999
1458	C1qtnf4	0.364	7.197	5.738
1605	Capzb	0.493	7.044	6.024
1679	Cbx3	2.458	5.980	7.278
1952	Cdc37	2.121	5.776	6.861
2018	Cdk5r1	0.485	6.346	5.304
2019	Cdk5r2	0.230	6.810	4.692
2204	Chmp2a	0.202	6.085	3.778
2224	Chrm1	0.450	6.542	5.389
2411	Cltb	0.371	6.666	5.234
2611	Cox6c	0.500	9.543	8.543
2703	Crip1	3.837	4.072	6.012
2704	Crip2	0.323	6.133	4.503
2748	Crym	0.341	6.174	4.622
2867	Ctxn1	0.446	8.172	7.006
2919	Cyb5a	0.391	6.354	4.999
3061	Dbp	0.142	6.430	3.618
3129	Ddt	2.060	6.045	7.088
3251	Dgcr6	0.204	6.053	3.756
3412	Dnal1	0.275	6.206	4.344
3546	Dusp1	0.496	6.005	4.992
3585	Dynlrb1	0.398	8.464	7.135
3650	Eef1b2	0.370	7.750	6.315
3715	Eif1b	0.467	6.969	5.870
3827	Enah	2.491	5.200	6.517
3834	Enho	0.293	6.306	4.536
3893	Epn1	0.402	6.721	5.406
3950	Esd	2.118	5.437	6.519
4117	Fam162a	0.015	6.579	0.526
4235	Fam96b	0.426	6.072	4.841
4296	Fbxo2	0.494	6.039	5.023
4392	Fgf12	2.317	6.152	7.364
4453	Fkbp1b	0.013	6.228	0.000
4454	Fkbp2	0.464	7.444	6.336
4455	Fkbp3	0.448	7.212	6.055
4586	Ftl1	0.451	7.956	6.808
4976	Gnb2l1	0.426	6.937	5.705
4984	Gng13	0.172	7.178	4.639
5010	Golga7	2.104	5.434	6.507
5233	Gsn	0.484	6.292	5.246
5252	Gstm7	0.455	7.131	5.996
5311	Guk1	0.467	7.643	6.544
5384	Hbb-b1	2.264	5.775	6.954
5504	Hist1h1d	0.352	7.387	5.880
5513	Hist1h2bh	0.233	7.106	5.003
5517	Hist1h4b	0.326	7.898	6.281
5518	Hist1h4m	0.194	8.378	6.009
5519	Hist2h2ab	0.057	6.688	2.553
5520	Hist2h3c2	0.267	7.400	5.494
5521	Hist2h4	0.176	9.279	6.775
5523	Hist3h2ba	0.400	6.036	4.713
5542	Hmga1	0.015	6.072	0.000
5660	Hrsp12	3.459	4.292	6.083
5762	Hypk	0.300	7.542	5.806
6076	Isca2	5.058	4.107	6.445
6123	Itm2a	2.477	4.937	6.245
6174	Junb	0.352	6.529	5.022
6209	Kcna1	2.036	5.150	6.176
6243	Kcnh5	2.314	4.869	6.079
6249	Kcnip3	0.371	6.164	4.733
6495	Kpna3	2.477	4.713	6.022
6577	LOC100134871	2.151	6.572	7.677
6647	LOC100911177	0.009	6.869	0.000
6684	LOC257642	125.852	0.012	6.988
6775	LOC498750	16.486	2.360	6.404
6916	LOC690871	0.012	6.419	0.000
6928	LOC691807	0.462	6.041	4.928
6941	Lage3	0.474	6.476	5.399
6960	Lamtor5	2.066	5.339	6.386
7371	Ly6h	0.419	6.584	5.328
7448	Maf1	2.240	4.983	6.146
7602	Matk	0.379	6.134	4.734
7841	Micu3	0.409	6.292	5.002
7874	Mir1188	2102.724	0.013	11.051
7886	Mir125b1	0.000	11.724	0.000
7910	Mir140	310.184	0.013	8.290
7945	Mir186	1655.080	0.013	10.706
8071	Mir341	0.001	9.469	0.000
8236	Mir6320	0.012	6.407	0.000
8302	Mir92b	0.000	11.066	0.000
8524	Mrpl49	0.326	6.064	4.448
8538	Mrps18a	2.377	5.349	6.598
8709	Myeov2	0.294	7.703	5.934
8732	Myl6	0.427	8.881	7.654
8734	Myl6l	92.838	0.012	6.549
8934	Ndufb2	0.194	7.882	5.516
8936	Ndufb4	0.473	8.456	7.376
8939	Ndufb7	0.160	6.986	4.347
8948	Ndufs6	3.382	5.180	6.938
8953	Ndufv3	0.368	6.415	4.974
8989	Nenf	0.171	6.373	3.828
9160	Nop10	4.964	3.925	6.237
11011	Pdcd4	2.046	5.473	6.506
11071	Pdp1	3.045	5.202	6.808
11111	Penk	0.241	6.059	4.004
11141	Pfdn2	0.318	7.026	5.375
11151	Pfn1	0.428	7.240	6.017
11235	Phyhipl	0.443	6.562	5.387
11297	Pink1	0.293	6.769	4.999
11321	Pja1	2.062	6.340	7.384
11406	Plekhb1	0.462	7.171	6.056
11492	Pnp	0.206	6.151	3.875
11541	Polr2f	0.424	6.607	5.368
11557	Polr3k	0.390	6.103	4.745
11754	Prdx5	0.434	7.919	6.714
11955	Psd	0.490	6.384	5.356
11973	Psma6	0.330	6.537	4.937
11979	Psmb2	2.202	5.106	6.245
11984	Psmb7	0.457	7.583	6.454
12335	RGD1559909	0.195	6.237	3.876
12620	Rac1	2.016	6.106	7.118
12669	Rap1b	0.395	6.121	4.780
12845	Rer1	4.671	3.863	6.087
12902	Rgs5	3.017	4.642	6.236
12965	Rims4	3.636	4.708	6.570
12991	Rmrp	0.407	15.748	14.450
12993	Rn18s	11049.639	0.014	13.445
12994	Rn28s	12992.748	0.014	13.679
12995	Rn45s	446.863	0.013	8.817
13133	Rpl12	0.271	8.339	6.458
13135	Rpl13a	0.449	9.562	8.409
13138	Rpl17	0.380	9.195	7.800
13147	Rpl24	0.445	9.357	8.188
13173	Rpl6	0.420	8.486	7.234
13178	Rplp0	0.474	8.150	7.073
13196	Rps11	0.471	8.742	7.656
13201	Rps15a	0.258	9.495	7.543
13202	Rps16	0.484	9.055	8.008
13203	Rps17	0.444	9.083	7.911
13236	Rps8	0.478	9.428	8.363
13335	S100a1	115.478	0.013	6.865
13338	S100a13	0.268	6.729	4.831
13390	Sarnp	2.091	4.950	6.013
13414	Sc5d	2.218	5.011	6.160
13424	Scand1	2.151	5.015	6.119
13479	Scrg1	2.620	5.975	7.364
13510	Sdhb	0.445	6.422	5.254
13774	Shfm1	0.465	6.357	5.253
13824	Sirt5	2.077	5.538	6.593
14346	Snrnp70	2.477	5.495	6.803
14353	Snrpd2	0.289	7.410	5.622
14356	Snrpg	0.173	6.435	3.902
14378	Snx2	3.748	4.544	6.450
14532	Spin1	0.490	6.256	5.226
14570	Sprn	2.530	4.993	6.332
14672	Sst	0.347	7.916	6.389
14678	Ssu72	0.342	6.391	4.845
14754	Stk24	3.248	4.554	6.254
14928	Synpr	0.464	6.243	5.136
15017	Taldo1	0.303	6.741	5.018
15096	Tbca	0.393	6.802	5.455
15142	Tceb2	0.470	7.953	6.864
15169	Tctex1d2	5.960	3.608	6.183
15478	Tmem14a	2.141	5.030	6.128
15738	Top1	2.171	5.030	6.149
16059	Ttc9b	0.101	7.037	3.726
16211	Ubl5	0.420	7.983	6.730
16309	Upf3a	2.202	4.924	6.063
16321	Uqcr11	0.445	8.937	7.770
16336	Use1	0.329	6.117	4.512
16411	Vamp1	2.064	6.238	7.283
16698	Vti1a	4.713	4.115	6.352
17307	mrpl11	3.192	4.399	6.074
17310	rnf141	9.136	3.528	6.720

## Data Availability

The data presented in this study are available on request from the corresponding author.

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
