# Peer review of "Regulation of Genes Related to Cognition after tDCS in an Intermittent Hypoxic Brain Injury Rat Model"

_genes, 2022, doi:10.3390/genes13101824_

Round 1

Reviewer 1 Report

The authors studied the effect of tDCS on cognition in a brain injury rat model and analyzed the changes of neurogenesis-related genes by using RNA sequence. However, the following issues need to be solved.

1. Extensive editing work is required.

2. Several parts are confusing. 1) In the MATERIALS AND METHODS: both 3 and 4 were displayed statistical analysis; 2) RESULTS: description of the results from Figure 2 is unclear; 3) RESULTS: authors demonstrated “…average of normalized read counts (RCs) >6 times) (Table 1)” in the part of “2. RNA sequencing data”, but no related information was supplied in Table 1.

3. In the whole manuscript, sample information used for gene expression was missing, such as sample type, number (Control and stimulation) etc.

4. Authors need to verify the sequencing results, at least the selected 12 genes.

5. Minor issues: 1) Figure 2 and 3 are unclear. 2) Table 1 should be formatted.

Author Response

The authors studied the effect of tDCS on cognition in a brain injury rat model and analyzed the changes of neurogenesis-related genes by using RNA sequence. However, the following issues need to be solved.

  1. Extensive editing work is required.

  [Author’s reply] Thank you for your valuable comment. We tried to edit this research extensively with English editing service.

  1. Several parts are confusing. 1) In the MATERIALS AND METHODS: both 3 and 4 were displayed statistical analysis; 2) RESULTS: description of the results from Figure 2 is unclear; 3) RESULTS: authors demonstrated “…average of normalized read counts (RCs) >6 times) (Table 1)” in the part of “2. RNA sequencing data”, but no related information was supplied in Table 1.

[Author’s reply] Thank you for your precise comment.

1) We merged the previous 3. Sample size and 4. Statistical analysis into 3. Statistical analysis. And we deleted overlapping significance level content.

2) We redraw the Figure 2 and explained more detail just as below;

“tDCS have an effect on cognitive function as demonstrated by statistically significant changes in escape latency and time differences between the stimulation and control groups.”

3) We added average of normalized read counts of each gene in Table1. And we included the whole RNA sequencing raw data as supplementary.

  1. In the whole manuscript, sample information used for gene expression was missing, such as sample type, number (Control and stimulation) etc.

[Author’s reply] Thank you for your precise comment. We included sample information used for gene expression in “Method Section” about the details just as below.

“After sacrifice, hippocampal tissues from all rats were extracted for RNA sequencing.”

  1. Authors need to verify the sequencing results, at least the selected 12 genes.

[Author’s reply] Thank you for your precise comment. We displayed sequencing results in Table 1. And we changed the title of the Table 1 to be more clearly. “Table 1. Fold changes of 180 genes that showed significant fold changes”. Selected 12 genes were sorted by KEGG mapper analysis.

  1. Minor issues: 1) Figure 2 and 3 are unclear. 2) Table 1 should be formatted.

[Author’s reply] Thank you for your precise comment. We modified figure 2 and 3 more detail. And we include the normalized data in Table 1.

===============================================

Thank you very much again.

Sincerely yours,

Min-Keun Song. MD, PhD

Department of Physical & Rehabilitation Medicine

Chonnam National University Medical School & Hospital

#42, Jebong-Ro, Dong-Gu, Gwangju, 61469, Republic of Korea

Telephone: + 82 62 220 5180 (Gwangju office)

FAX: + 82 62 228 5975 (Gwangju office)

Reviewer 2 Report

The manuscript was well written and explored the effect of tDCS on cognition from the perspective of gene expression by microarray technique. The author needs to justify this curiosity:

1.  Why were the control rats without any treatment and tDCS alone not performed?  

2. When precisely was the time the tDCS was applied to the rats in Hypoxic injury? Unclear explanation in text. Could be after 12 hours in a hypoxic chamber.  

3, Could the MWM test have been performed after seven days of tDCS stimulation? Was this time the platform taken out or just hidden in the pool?      

Author Response

The manuscript was well written and explored the effect of tDCS on cognition from the perspective of gene expression by microarray technique. The author needs to justify this curiosity:

  1. Why were the control rats without any treatment and tDCS alone not performed?  

 [Author’s reply] Thank you for your valuable comment. We would like to compare with spontaneous recovery and tDCS stimulation. The control rats were exposed at normal oxygen concentration. However, The sham was not applied. We included the purpose of the control rats without any treatment in the Method section and lack of sham group for tDCS in the Limitation section.

  1. When precisely was the time the tDCS was applied to the rats in Hypoxic injury? Unclear explanation in text. Could be after 12 hours in a hypoxic chamber.  

[Author’s reply] Thank you for your precise comment. We explained the detailed time sequence and put the time table just as below in “Materials and Method section”

3, Could the MWM test have been performed after seven days of tDCS stimulation? Was this time the platform taken out or just hidden in the pool?   

 [Author’s reply] Thank you for your precise comment. We performed the MWM test after seven days of tDCS. We measured the escape latencies to put the platform below the surface of water. We include the sentence just as below in “Method Section”

“The rats tried to find the platform below the surface of water within 300 seconds.” 

================================================

Thank you very much again.

Sincerely yours,

Min-Keun Song. MD, PhD

Department of Physical & Rehabilitation Medicine

Chonnam National University Medical School & Hospital

#42, Jebong-Ro, Dong-Gu, Gwangju, 61469, Republic of Korea

Telephone: + 82 62 220 5180 (Gwangju office)

FAX: + 82 62 228 5975 (Gwangju office)

Round 2

Reviewer 1 Report

Although the authors demonstrated “Selected 12 genes were sorted by KEGG mapper analysis” in response to the issue “Authors need to verify the sequencing results, at least the selected 12 genes”, the sequencing results are still needed to be verified at gene or/and protein level. Otherwise, the present data are not sufficient to support the conclusion.

Author Response

Although the authors demonstrated “Selected 12 genes were sorted by KEGG mapper analysis” in response to the issue “Authors need to verify the sequencing results, at least the selected 12 genes”, the sequencing results are still needed to be verified at gene or/and protein level. Otherwise, the present data are not sufficient to support the conclusion..

[Author’s reply] Thank you for your valuable comment. We did not perform the quantitative data such as real-time PCR or Western blot analysis. This weak point was written in Limitation part just as below;

“the lack of quantitative data for selected genes, such as real-time PCR or Western blot analysis, which was not supported.”

================================================

Thank you very much again.

Sincerely yours,

Min-Keun Song. MD, PhD

Department of Physical & Rehabilitation Medicine

Chonnam National University Medical School & Hospital

#42, Jebong-Ro, Dong-Gu, Gwangju, 61469, Republic of Korea

Telephone: + 82 62 220 5186

FAX: + 82 62 228 5975
